# Addressing Myopic Constrained POMDP Planning
# with Recursive Dual Ascent

**Primary Keywords:** *None*

## Abstract

Lagrangian-guided Monte Carlo tree search with global dual ascent has been applied to solve large constrained partially observable Markov decision processes (CPOMDPs) online. In this work, we demonstrate that these global dual parameters can lead to myopic action selection during exploration, ultimately leading to suboptimal decision making. To address this, we introduce history-dependent dual variables that guide local action selection and are optimized with recursive dual ascent. We empirically compare the performance of our approach on a motivating toy example and two large CPOMDPs, demonstrating improved exploration, and ultimately, safer outcomes.

## Introduction

Deploying autonomous systems in complex environments requires methods for safe planning under uncertainty. Constrained partially observable Markov decision processes (CPOMDPs) are mathematical models that codify decision making problems whose solutions must satisfy cost constraints under both outcome and state uncertainty (Kochenderfer, Wheeler, and Wray 2022; Kim et al. 2011; Poupart et al. 2015). From robotics systems (Kurniawati 2022) to geological carbon sequestration (Corso et al. 2022), the CPOMDP framework enables safe policy generation in a diverse set of applications entailing uncertainty.

Offline solutions find approximately optimal policies but are limited to small discrete state, action, and observation spaces. Online methods can scale to larger spaces by searching across reachable outcomes. For example, cost-constrained partially observable Monte Carlo tree search (CC-POMCP), searches for safe actions online using a Lagrangian-guided heuristic that trades off future costs and rewards through dual variables $\lambda$ (Lee et al. 2018). Crucially, a single set of *global* dual variables are optimized by dual ascent and shared between all history nodes during search.

Lagrangian-guided action selection with global dual variables can lead to myopic decision making. Consider the simple CPOMDP depicted in Figure 1, where an agent must maximize a terminal reward $R$ while satisfying a cost $C \leq 1$. The optimal strategy takes $a_1$ followed by $a_2$ towards a belief $b_4$ that yields maximal reward while satisfying the cost constraint. However, guiding search with a global dual parameter will fail to explore the optimal action sequence,

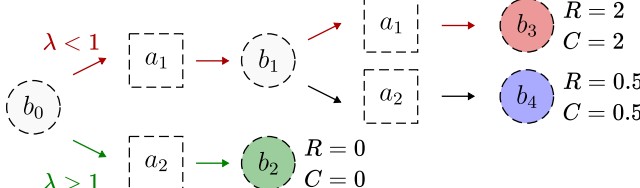

Figure 1: A CPOMDP illustrating myopic decision making when guiding search with global dual variables. With a global $\lambda$, search explores either the cautious green belief or the budget violating red belief and misses the optimal, feasible blue belief.

as large dual variables will explore the cautious low-reward node $b_2$ while small dual variables will explore the infeasible high-reward node $b_3$. Exploring the optimal node requires a different strategy after taking the first $a_1$ action.

To address this problem, we augment Lagrangian-guided Monte Carlo tree search (MCTS) (Silver and Veness 2010) algorithms with history-dependent, *local* dual variables. These local variables are optimized recursively by dual ascent at each history node against local constraint violations. By adapting safe action selection to local constraint violations in different beliefs, we can better explore optimal safe paths, ultimately yielding better policies.

To evaluate our approach, which we denote by appending a '+' to the underlying algorithms, we conduct experiments in both discrete and continuous spaces on an illustrative small problem and two larger CPOMDPs. Results demonstrate that constrained MCTS with global dual parameters can lead to either excessively risky or cautious exploration, issues our modifications address.

In summary, our main contributions are to (i) highlight myopic decision making in CPOMDPs, (ii) propose recursive dual ascent with history-dependent dual variables to improve online dual-ascent-guided CPOMDP planning, and (iii) empirically demonstrate improved search efficiency and constraint satisfaction in three problem domains.

## Background

**CPOMDPs** CPOMDPs provide a structured approach for optimizing decision making under uncertainty when considering competing objectives (Roijers et al.

2013). A CPOMDP can be defined with the tuple $\langle \mathcal{S}, \mathcal{A}, \mathcal{O}, T, Z, R, \mathbf{C}, \hat{\mathbf{c}}, \gamma, b_0 \rangle$. At each step, an agent in a partially observable state $s \in \mathcal{S}$ takes an action $a \in \mathcal{A}$, transitions to a successor state, and emits an observation $o \in \mathcal{O}$. Transitions and observations follow Markov distributions $T$ and $Z$, which model outcome uncertainty and partial observability. Each transition emits an immediate reward following $R$ and $k$ immediate costs following $\mathbf{C}$. Optimal CPOMDP planning selects actions to maximize expected cumulative rewards while satisfying expected cumulative cost budgets $\hat{\mathbf{c}}$.

As states are partially observable, agents must plan using histories of past actions and observations $h_t$, which may be succinctly represented as instantaneous beliefs over state $b_t = b(s_t) = Pr(s_t = s \mid h_t)$. In infinite horizons, CPOMDP policies $\pi$ optimize expected *discounted* cumulative reward $V_R^\pi$ from an initial belief $b_0$ such that expected discounted costs $\mathbf{V}_C^\pi$ satisfy their budgets in expectation:

$$\max_\pi V_R^\pi(b_0) = \mathbb{E}_\pi \left[ \sum_{t=0}^\infty \gamma^t R(b_t, a_t) \mid b_0 \right] \quad (1)$$

$$\text{s.t. } V_{C_k}^\pi(b_0) = \mathbb{E}_\pi \left[ \sum_{t=0}^\infty \gamma^t C_k(b_t, a_t) \mid b_0 \right] \leq \hat{c}_k \; \forall \, k,$$

where $\gamma \in (0, 1)$ is a discount factor that bounds objectives. This constrained optimization problem can be expressed equivalently through its Lagrangian

$$\max_\pi \min_{\boldsymbol{\lambda} \geq 0} \left[ V_R^\pi(b_0) - \boldsymbol{\lambda}^\top (\mathbf{V}_C^\pi(b_0) - \hat{\mathbf{c}}) \right], \quad (2)$$

where $\boldsymbol{\lambda}$ are dual variables.

**Online Planning in CPOMDPs** Offline solution methods to CPOMDPs, such as point-based backups (Kim et al. 2011), approximate linear programming (Poupart et al. 2015), column generation (Walraven and Spaan 2018), and projected gradient ascent (Wray and Czuprynski 2022), cannot be applied to continuous or large problems. In contrast, online planning can scale to large state spaces by considering the set of reachable histories at runtime.

Online solution methods have coupled search with safety critics from offline solutions (Undurti and How 2010) or previous searches (Parthasarathy et al. 2023). Rather than rely on a critic, cost-constrained partially observable Monte Carlo planning (CC-POMCP) (Lee et al. 2018) optimizes Equation (1) online directly by interleaving a Lagrangian-guided partially observable MCTS with dual ascent. In MCTS, actions and their resulting transitions are sampled to gradually build a search tree that estimates reward and cost values.

In CC-POMCP exploration, actions are chosen at each history node $h$ to maximize

$$\mathcal{Q}_{\boldsymbol{\lambda}}^\oplus(h, a) = \left[ Q_R(h, a) + \boldsymbol{\lambda}^\top \mathbf{Q}_C(h, a) + \kappa \sqrt{\frac{\log N(h)}{N(h, a)}} \right], \quad (3)$$

where $Q_R$ and $\mathbf{Q}_C$ are reward and cost action value estimates, $\boldsymbol{\lambda}$ are dual variable estimates, and node visitation

counts $N$ are used for optimistic exploration (Auer, Cesa-Bianchi, and Fischer 2002).

In between search simulations, dual variable estimates are updated by gradient descent on Equation (3), updating away from cost violations at the root node.[1] Crucially, though these updates aim to satisfy constraints at the root node, the same dual variables are then used globally for action selection in subsequent decisions. To extend CC-POMCP to continuous action and observation spaces, the constrained partially observable Monte Carlo planning with observation widening (CPOMCPOW) algorithm uses double progressive widening to artificially limit branching factors based on node visit counts (Jamgochian, Corso, and Kochenderfer 2023).

## Approach

Lagrangian-guided action selection with global dual variables as presented in Equation (3) can lead to myopic decision making. Consider the simple CPOMDP depicted in Figure 1, where an agent must maximize reward while satisfying a cost budget $\hat{c} = 1$. Guiding search with a global dual parameter will fail to explore the optimal action sequence, because exploration of the optimal node would necessitate a different strategy than used to take the first action.

To address this issue, we propose augmenting Lagrangian-guided tree search algorithms to enable history-dependent constrained exploration. In our augmentation, each history node maintains its own *local* dual variables $\boldsymbol{\lambda}(h)$ used to guide local action selection. These dual variables are optimized separately using a recursive dual ascent that updates them away from constraint violations in their subtrees. We refer to our augmentations by adding a '+' to the underlying algorithms.

Algorithm 1 depicts our adaptation applied to CC-POMCP, with differences to the underlying algorithm highlighted in blue. The SIMULATE procedure of a constrained MCTS algorithm recursively simulates trajectories to a fixed depth and backpropagates rewards and costs to iteratively build a search tree. A ROLLOUT, where the algorithm runs an estimate policy to completion, provides an initial value approximation when leaf nodes are first encountered. In our augmentation, leaf nodes also initialize local dual variables with the values of their parents (line 13). Actions are then selected using local dual variable estimates (line 16), with optimism guided by Equation (1) (line 27) and a stochastic policy of best actions formed by the STOCHASTICPOLICY method of Lee et al. (2018).

While using backpropagated rewards and costs to update value and single-step cost estimates (lines 21–23), our augmentation performs dual ascent to guide local dual variables to penalize constraint violations in their associated subtrees (line 24). Doing so requires forward propagating cost estimates for earlier actions to estimate the remaining budget in each subtree $\hat{c}_{\text{rem}}$. This is done by using the single-step cost estimates $\bar{\mathbf{c}}$ to estimate $\hat{\mathbf{c}}(hao) = (\hat{\mathbf{c}}(h) - \bar{\mathbf{c}}(ha))/\gamma$ (line 21). This recursive dual ascent procedure enables history

---

[1]The term dual *ascent* arises from convention that minimizes the objective rather than maximizing it.

**Algorithm 1: CC-POMCP+**

---

1: **procedure** PLAN($h$)
2:     $\boldsymbol{\lambda} \leftarrow \boldsymbol{\lambda}_0$
3:     **for** $i \in 1 : n$
4:         $s \leftarrow$ sample from $b$
5:         SIMULATE($s, h, \hat{\mathbf{c}}, d_{\max}$)
6:         $a \sim$ UCBPOLICY($h, \boldsymbol{\lambda}, 0, 0$)
7:         $\boldsymbol{\lambda} \leftarrow [\boldsymbol{\lambda} + \alpha_i(\mathbf{Q}_C(h, a) - \hat{\mathbf{c}})]$
8:     **return** UCBPOLICY($h, \boldsymbol{\lambda}, 0, \nu$)
9: **procedure** SIMULATE($s, h, \hat{\mathbf{c}}_{\text{rem}}, d$)
10:     **if** $d = 0$
11:         **return** $[0, \mathbf{0}]$
12:     **if** $h \notin T$
13:         $\boldsymbol{\lambda}(h) \leftarrow \boldsymbol{\lambda}(h \setminus a^- o^-)$
14:         $T(ha) \leftarrow (N_{\text{init}}, Q_{R,\text{init}}, \mathbf{Q}_{C,\text{init}}, \bar{\mathbf{c}}_{\text{init}}) \forall a$
15:         **return** ROLLOUT($s, h, \hat{\mathbf{c}}_{\text{rem}}, d$)
16:     $a \sim$ UCBPOLICY($h, \boldsymbol{\lambda}(h), \kappa, \nu$)
17:     $s', o, r, \mathbf{c} \leftarrow G(s, a)$
18:     $[R, \mathbf{C}] \leftarrow [r, \mathbf{c}] +$
            $\gamma \cdot$ SIMULATE($s', hao, \frac{\hat{\mathbf{c}}_{\text{rem}} - \bar{\mathbf{c}}(ha)}{\gamma}, d - 1$)
19:     $Q_R(ha) \leftarrow Q_R(ha) + \frac{R - Q(ha)}{N(ha)}$
20:     $N(h) \leftarrow N(h) + 1$
21:     $N(ha) \leftarrow N(ha) + 1$
22:     $\bar{\mathbf{c}}(ha) \leftarrow \bar{\mathbf{c}}(ha) + \frac{\mathbf{c} - \bar{\mathbf{c}}(ha)}{N(ha)}$
23:     $\mathbf{Q}_C(ha) \leftarrow \mathbf{Q}_C(ha) + \frac{\mathbf{C} - \mathbf{Q}_C(ha)}{N(ha)}$
24:     $\boldsymbol{\lambda}(h) \leftarrow [\boldsymbol{\lambda}(h) + \alpha_{N(h)}(\mathbf{Q}_C(ha) - \hat{\mathbf{c}}_{\text{rem}})]^+$
25:     **return** $[R, \mathbf{C}]$
26: **procedure** UCBPOLICY($h, \boldsymbol{\lambda}, \kappa, \nu$)
27:     $\mathcal{Q}_{\boldsymbol{\lambda}}^{\oplus}(h, a) =$
        $\left[ Q_R(h, a) + \boldsymbol{\lambda}^\top \mathbf{Q}_C(h, a) + \kappa \sqrt{\frac{\log N(h)}{N(h, a)}} \right]$
28:     **return** STOCHASTICPOLICY($\mathcal{Q}_{\boldsymbol{\lambda}}^{\oplus}, \nu$)

---

nodes to adapt their exploration strategies based on local constraint violations in their subtrees. This results in more accurate value estimation, as future nodes can make decisions appropriate to their local constraint violations.

As with CC-POMCP, planning at each step entails interweaving simulations (line 5) with dual ascent at the root node (line 7). After executing the best root node policy, planning at the next step necessitates updating the belief using new observations and the remaining cost budget using the immediate costs associated with the executed actions, which can be estimated using $\bar{\mathbf{c}}$. Though not shown, our augmentations extend trivially to CPOMCPOW, which uses progressive widening to overcome the multitude of branching in continuous problems. We note that our method could also be used to plan online in recursively constrained POMDPs (RC-POMDPs), which overcome inconsistencies of Bellman optimality in CPOMDPs through the recursive application of constraints (Ho et al. 2023).

## Experiments

In this section, we compare the overall performance and search efficacy with and without local dual parameters in three CPOMDPs of varying sizes. Our experiments are performed in Julia using the POMDPs.jl framework (Egorov et al. 2017). Anonymized code with implementation details is available for peer review and will be released upon publication.

**CPOMDP Problems** We first briefly overview our target problem domains, denoting whether their state, action, and observation spaces are discrete (D) or continuous (C):

1. **Constrained Tiger** (D, D, D) Adapted from Kaelbling, Littman, and Cassandra (1998), in this CPOMDP environment high reward comes with high cost. The agent must avoid a tiger who is behind either the left or right door by either noisily listening to localize the tiger or opening a door.
2. **Constrained LightDark** (C, D, C): In this CPOMDP from Lee et al. (2018), an agent must localize itself before navigating to a goal without entering a high cost region.
3. **Constrained Spillpoint** (C, D, C): Introduced as a POMDP by Corso et al. (2022) to guide safe carbon sequestration within subsurface formations under geological uncertainty. As done by Jamgochian, Corso, and Kochenderfer (2023), we replace penalization for $CO_2$ leaks with constraint violation.

**Search Efficacy** Generating accurate action value estimates within a Monte Carlo search tree requires repeated node visits. It is therefore desirable for an online solver to spend most of its planning budget simulating the highest reward nodes that satisfy constraints. We first see that our modifications can improve search efficacy by spending more time exploring optimal action sequences.

| Model | $N(b_0, a)/N(b_0)$ | $Pr(a \mid b_0)$ |
|---|---|---|
| CPOMCPOW+ | [0.34, **0.63**, 0.02] | [0.00, **0.82**, 0.00] |
| CPOMCPOW | [0.08, 0.49, 0.09] | [0.05, 0.65, 0.09] |

Table 1: Statistics comparing the exploration and execution of initial actions $+1, +5, +10$ in Constrained LightDark across all runs. CPOMCPOW+ concentrated search on the optimal action for a larger fraction of the $6 \times 10^5$ search queries, and ultimately chose $+5$ in more trials.

For example, in the Constrained LightDark CPOMDP, $a = +5$ is the best initial step the agent can take without violating the constraint above $s = 12$. In Table 1, we see that the modified solver concentrates more of the search around $a = +5$ despite it having subtrees with both unsafe and safe subsequent actions. This ultimately results in more trials that select the optimal action.

We see similar behavior in the Constrained Tiger CPOMDP. To visualize this, Figure 2 compares key areas of the converged search trees for the first two actions. On the right is CC-POMCP with global dual variables and on the left is our proposed modification with local dual variables.

In the CC-POMCP solution, the root node global dual variable $\boldsymbol{\lambda}_g$ converges low, confining the search to actions

| Domain | Simulations | $|S|, |A|, |O|, \gamma$ | $\hat{\mathbf{c}}$ | Algorithm | Reward, $\hat{V}_R$ | Cost, $\hat{V}_C$ | Violations (%) |
|---|---|---|---|---|---|---|---|
| Constrained Tiger | 100 | $2, 3, 2, 0.95$ | 0.9 | CC-POMCP | $-4.83_{\pm 0.82}$ | $0.66_{\pm 0.04}$ | 59 |
| | | | | CC-POMCP+ | $-16.17_{\pm 0.79}$ | $0.30_{\pm 0.04}$ | **27** |
| Constrained LightDark | 100 | $\infty, 7, \infty, 0.95$ | 0.1 | CPOMCPOW | $30.31_{\pm 7.56}$ | $0.092_{\pm 0.027}$ | 10 |
| | | | | CPOMCPOW+ | $28.00_{\pm 7.47}$ | $0.017_{\pm 0.012}$ | **2** |
| Constrained Spillpoint | 10 | $\infty, 20, \infty, 0.9$ | 0.0 | CPOMCPOW | $3.90_{\pm 0.55}$ | $0.001_{\pm 0.000}$ | **80** |
| | | | | CPOMCPOW+ | $3.93_{\pm 0.55}$ | $0.001_{\pm 0.000}$ | 90 |

Table 2: Performance with and without local dual variables on three problem domains comparing mean and standard error of discounted cumulative rewards and costs alongside the fraction of failed runs. Lowest constraint violation rates are bolded.

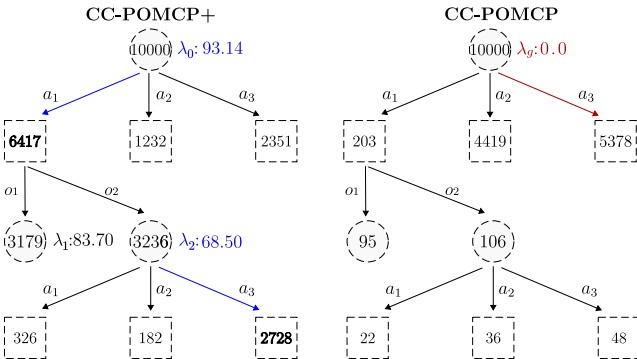

Figure 2: Constrained Tiger history trees showing visitation count $N$ with local versus global dual variables. Left to right actions are listen noisily, open left door, and open right door; and observations are tiger heard behind the right door and tiger *not* heard behind the right door. Local dual variables enable better exploration of the optimal action path, which necessitates listening before selecting actions.

with high cost and high reward. As a result, the agent immediately opens a door, often incurring an immediate cost violation. In the CC-POMCP+ solution, the root node dual variable converges higher than the subsequent dual variables, allowing the search to explore low cost low reward nodes that listen for the tiger before safely opening a door. This safe exploration is reflected by the larger visitation counts for safe actions.

**Performance** Table 2 compares mean rewards and costs on the three problems; lower rewards that satisfy cost constraints are preferred over violating constraints. In Constrained LightDark, CPOMCPOW+ achieves similar average rewards as CPOMCPOW without violating the constraint limit, as seen in Figure 3. CPOMCPOW+ is more likely to explore actions at the safety limit because it can avoid subsequent high costs after taking risky initial steps.

In contrast, CPOMCPOW+ had similar performance as the original CPOMCPOW when run on Constrained Spillpoint. In problems with small cost budgets, when no cost violation is allowable, a high-value global dual variable suffices to safely guide action selection. When run on Constrained Tiger, the modified algorithm obtained lower discounted total rewards but halved cost violations when over half of trials without the modification violated constraints.

Our modifications adapt Lagrangian-guided tree search

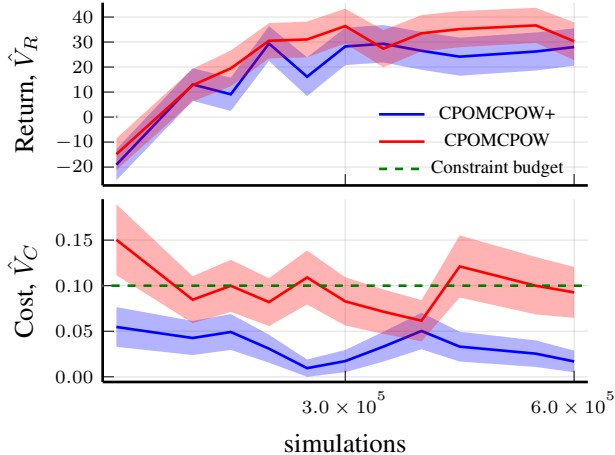

Figure 3: Discounted results for Constrained LightDark. Rewards increase comparably for both methods.

with history-dependent dual variables. The resulting fine-tuned algorithm guides expected action values to converge to the best outcome within their control. The final decision is made with improved information and therefore leads to improved outcomes.

## Conclusion

Constrained POMDP solvers optimize planning objectives while satisfying cost constraints under state and outcome uncertainty. Constrained Monte Carlo tree search methods guided by global dual variables can be myopic, yielding excessively risky or cautious actions as a result of improper exploration. To address this, we proposed local dual variables to guide safe, history-dependent action selection, and optimized them using recursive dual ascent. We empirically demonstrated that our approach resulted in a decrease in simulations spent exploring unsafe or overly cautious actions, ultimately improving safe outcomes.

**Limitations** Our modification only benefits problems that require adapting safe exploration at different beliefs. Additionally, our modifications still inherit the other shortcomings of their underlying algorithms, namely, lack of anytime safety guarantees and high variance in costs and returns.

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
