# OpenReview forum: "Addressing Myopic Constrained POMDP Planning with Recursive Dual Ascent"
_icaps-conference.org/ICAPS/2024/Conference — ICAPS 2024_

### Official Review · Reviewer_bv6W · 2024-01-21

**Significance And Importance:** 2
**Soundness:** 3
**Novelty:** 2
**Clarity:** 4
**Overall Evaluation:** 2
**Confidence:** 5

**Weaknesses:**

1: Minor weaknesses that are easily fixable.

**Contributions Of The Paper:**

This paper discusses Lagrangian-guided Monte Carlo tree search using dual ascent for constrained POMDPs, and show that global dual ascent can lead to myopic decision making. Since the dual variable effectively works as a weighting between rewards and costs, when the dual variables are fixed throughout the tree, exploration of the optimal node further down the tree may fail or perform badly since the optimal weighting between rewards and costs may need to change in different parts of the tree. To resolve this issue, this paper proposes using history-dependent local dual variables that are optimized recursively against local cost constraints via dual ascent. Empirical results show that this leads to safer action selections and yields better policies that are safer.

**Ethical Considerations:**

(1) Not Applicable: The paper does not have any ethical considerations to address

**Nomination For Best Paper:**

No

**Questions For Authors:**

1. Does the augmentation of local dual variables affect the asymptotic convergence and optimality of the CPOMCP(OW) algorithm for CPOMDPs?

2. The constraint violation rate is not defined. Is it the number of runs that violate the cost constraints at the root divided by the total number of runs?


POST REBUTTAL
Thank you for your response. My questions have been sufficiently answered.

**Reproducibility:**

5: Code and domains (whichever apply) are already publicly available

**Strengths Of The Paper:**

1. This paper is well written and clear, and is a good (albeit small) contribution to the literature on CPOMDP planning.

2. The use of local dual variables in the tree is well motivated and is a cool idea to enable different “weighting" between rewards and costs at different parts of the tree depending on how close to constraint violation a subtree is.

3. The empirical results show its usefulness in computing safer policies with less constraint violations while being competitive in expected rewards.

Overall, this is an interesting method to resolve some drawbacks with current methods in the literature for CPOMDPs, and I recommend an accept as a short paper.

**Weaknesses Of The Paper:**

I don't have much to write in terms of major weaknesses because I find it well written for the scope of the paper.

1. Although the motivation and empirical results are convincing, I would have liked if there was more discussion on how the local history-dependent dual variables may impact (rate of) convergence and optimality of the algorithm. It is not directly clear to me that if each local dual variables converge to its own optimal value, that the tree search would (asymptotically) converge to the optimal value for the original problem of Eq. 1.

2. I believe a citation for the Lagrangian form (Eq. 2) of the constrained optimization problem is warranted.

3. The constraint violation rate is not well defined.

---

> ### Author Rebuttal · Authors · 2024-01-28
>
> Thank you for your review. We will address the errors in reference formatting and include the missing citation for the Lagrangian form (Eq. 2).
>
> 1. Does the augmentation of local dual variables affect the asymptotic convergence and optimality of the CPOMCP(OW) algorithm for CPOMDPs?
>
> This is correct. Each lambda variable would need to converge on its own. While this approach may increase the required time to convergence because some subtrees are explored less often, we note a new node’s lambda value’s convergence time is theoretically reduced by its initialization; we use its parent value in line 13 of Algorithm 1. However, the benefit of using local dual variables may not be apparent at lower iteration numbers. Previous work on CMCTS stipulates that MCTS convergence to an optimal policy depends on exploration being ‘appropriately controlled’ [3]. Search must cover the full problem space to find the best nodes and then concentrate on the best nodes to increase certainty in their value. We argue that local dual variables properly guide search among subtrees. Consequently, convergence may take longer but will converge to the optimal safe path.
>
> 2. The constraint violation rate is not defined. Is it the number of runs that violate the cost constraints at the root divided by the total number of runs?
>
> The constraint violation rate is the number of runs that violate the cost budget by the end of the run. Therefore in Table 2, Constrained Tiger with global dual variables has 100 total runs and 59 of them had total discounted costs that violated the cost budget = 0.9. This does not mean that 59 of the runs violated the full cost budget at the first step, but that they violated the total allowable budget by the end of the run. Figure 2 shows an example tree where the budget was violated at the initial step, but this is not the case for all runs of all problems.
>
> [3] Silver, D.; and Veness,  J. 2010. Monte-Carlo planning in large POMDPs. In Advances in Neural Information Processing Systems (NeurIPS), 2164–2172.

---

### Official Review · Reviewer_euzg · 2024-01-22

**Significance And Importance:** 2
**Soundness:** 3
**Novelty:** 2
**Clarity:** 3
**Overall Evaluation:** 1
**Confidence:** 3

**Weaknesses:**

1: Minor weaknesses that are easily fixable.

**Contributions Of The Paper:**

This paper looks at on-line planning for constrained POMDPs using Monte Carlo tree search.
Previous work (CC-POMCP and CPOMCPOW) relied on going through the Lagrangian relaxation of the constrained problem to include the cost constraints in the value estimates of each node, but using the same Lagrangian dual variable $\lambda$ (optimized online) across the whole tree. This approach leads to poor behaviors in some scenarios, and thus does not guarantee converging to optimal decisions in the limit.

The present paper proposes variants of both aforementionned algorithms wherein a different Lagrangian dual variable $\lambda(h)$ is used in each node (each attached to an observation-action history).
Experiments are conducted on three classical benchmark problems, showing the benefit of the approach when enough iterations can be performed to start learning the new variables. Otherwise, sticking to the "single-dual variable" approach can be preferable. The exploration behavior is also examined, for instance through comparing history trees on the constrained Tiger problem.

**Ethical Considerations:**

(1) Not Applicable: The paper does not have any ethical considerations to address

**Nomination For Best Paper:**

No

**Questions For Authors:**

[POST-REBUTTAL COMMENT]
Let me just thank the authors for their responses.
I have no further comment at this stage.


1. How does StochasticPolicy() work?
   What is $\nu$?

2. How is $\nu$ set in Algorithm 1?

3. Any comment about weights $\alpha_i$? How is it defined in the experiments? Is it as in previous work?

**Reproducibility:**

4: Authors promise to release code and domains (whichever apply).

**Strengths Of The Paper:**

IMHO, deriving good online solvers for constrained POMDPs is an interesting problem.
The paper identifies a key weakness of existing approaches in this field, proposes a solution, and presents an empirical evaluation that validates its claims and points out limitations (the need for a sufficient number of iterations).

Overall the paper is well organized, well written and clear. I did not find clear technical issues.

**Weaknesses Of The Paper:**

The key idea behind the contribution is fairly simple in the end, but it is a natural step to follow to address a weakness of the state of the art.

As far as I understand, the paper requires some familiarity with the state-of-the-art to properly understand why CC-POMCP's exploration behavior fails. The explanations are too "high-level" for me.  It seems to me that one needs to understand what the StochasticPolicy() function does.
In Algorithm 1, I don't see where $\nu$ is set (and don't know what is its purpose since its linked to StochasticPolicy()).

It would seem important to me to discuss the theoretical guarantees of the proposed approach, whatever they are.
[As far as I remember, Bellman's optimality principle does not apply to C-POMDPs (there are temporal inconsistencies), so that a POMCP-based algorithm cannot come with theoretical guarantees.]



Other comments (not really weaknesses):
--------------

- Table captions are usually above the table itself.

- "... converges low" and "... converges higher" are unusual to me.
  But that still may be proper English.

- In Table 2, I would suggest adding intermediate horizontal lines between problems, maybe using \cmidrule{2-8} (if using booktabs).

<- "to explore low cost low reward nodes"
-> "to explore low-cost low-reward nodes" ??

- [References]: There is no "in" field in (Ho et al., 2023).

- [References]: There is no "in" field in (Parthasarathy et al., 2023).

- [References]: There are too many capital letters in the title of (Kurniawati, 2022).

---

> ### Author Rebuttal · Authors · 2024-01-28
>
> Thank you for your review. We will improve the readability of Table 2 and address the noted grammatical errors.
>
> 1. How does StochasticPolicy() work? What is $\nu$? 2. How is $\nu$ set in Algorithm 1?
>
> The StochasticPolicy method [2] accounts for the error in estimating the scalarized action value, the Lagrangian action value $Q_{\lambda}$, when using it to select a best action. The policy uses $\nu$ to choose a set of Lagrangian action values within some uncertainty to the best Lagrangian action value, then attempts to select an optimized safe action from that set. In our experiments, we effectively set $\nu$ to 0, such that the action with the absolute highest Lagrangian action value is selected. In the case where multiple actions share the exact same highest Lagrangian action value, our method selects between them with equal probability. Because it belongs to the general body of CMCTS work, we now directly reference the original paper.
>
> 3. Any comment about weights, $\alpha_{i}$? How is it defined in the experiments? Is it as in previous work?
>
> In both Constrained LightDark and Constrained Spillpoint we used the same constant alpha weights, meaning $\alpha_{i}$ is always the same and also equal to $\alpha_{N(h)}$, as in previous work. This was chosen to improve the direct comparability between the modified and unmodified algorithms, though using  different alpha values is not necessarily invalidating as these methods require parameter tuning.
>
> The smaller problem Constrained Tiger also uses a constant alpha schedule. We chose the value of alpha and the maximum number of iterations together to ensure that the root node lambda value had stabilized.
>
> [2] Lee, J.; Kim, G.-H.; Poupart, P.; and Kim, K.-E. 2018. Monte-Carlo tree search for constrained POMDPs. In Advances in Neural Information Processing Systems (NeurIPS), volume31.

---

### Official Review · Reviewer_jTCm · 2024-01-22

**Significance And Importance:** 2
**Soundness:** 3
**Novelty:** 3
**Clarity:** 3
**Overall Evaluation:** 2
**Confidence:** 2

**Weaknesses:**

2: No major or minor weaknesses.

**Contributions Of The Paper:**

This short paper deals with the problem of using global dual parameters in Lagrangian-guided MCTS to solve large CPOMDPs online. The specific issue this paper focuses on is the myopic action selection leading to suboptimal solutions. To alleviate this issue, the paper proposes to use history-dependent, local dual variables that are optimized recursively at each history node against local constraint violations. The results show improved performance on two large CPOMDPs.

**Ethical Considerations:**

(1) Not Applicable: The paper does not have any ethical considerations to address

**Nomination For Best Paper:**

No

**Questions For Authors:**

1. Why does the modification only benefit problems that require adapting safe exploration at different beliefs?

2. What happens if the solution that conforms to the given constraint/budget does not exist?

3. What is $hao$ in line 169?

4. Shouldn’t the probabilities in the last column of Table 1 sum to 1? I am not sure if I am missing something.

**Reproducibility:**

4: Authors promise to release code and domains (whichever apply).

**Strengths Of The Paper:**

1. The paper introduces a simple yet useful solution of using history-dependent dual variables that solves a specific problem of avoiding myopic action selection in solving CPOMDPs.

2. The explanation is clear, and the presentation of changes in the CC-POMCP algorithm to create the CC-POMCP+ algorithm makes it very easy to understand the updates.

**Weaknesses Of The Paper:**

1. The search efficacy section explaining the choice of action for constrained light-dark CPOMDPP can be improved. It is not directly clear why a=+5 is the best initial action. I understand the space is limited, but adding a few sentences to improve the example would go a long way in improving the readability. The description of CPOMDP Problems can be reduced to make space if needed.

2. Shouldn’t the probabilities in the last column of Table 1 sum to 1? I am not sure if I am missing something.

---

> ### Author Rebuttal · Authors · 2024-01-28
>
> Thank you for your review. The search efficacy section, Line 223, can be edited for clarity: “the largest initial step the agent may take towards the light region to localize itself is +5. Larger steps are not ensured to be safe because of the uncertainty in the agent’s initial location.”
>
> 1. Why does the modification only benefit problems that require adapting safe exploration at different beliefs?
>
> This modification locally guides search with unique dual variables. However, if the same action always results in an unacceptable cost, for example it carries an extremely high cost regardless of belief and no cost is allowable (as in the case of Constrained Spillpoint), then universally high global dual variables will penalize that action everywhere.
>
> 2. What happens if the solution that conforms to the given constraint/budget does not exist?
>
> Because any cost budget can be set, it is certainly possible to make the problem infeasible. Previous work has suggested that if the resulting policy’s expected total cost exceeds the budget, a possible solution can be found by conducting a binary search for an ‘artificial bound’ cost that is obeyed [1]. Although our paper does not explicitly cover this scenario, larger constraint violations, even if inevitable, would favor a policy to minimize cost.
>
> 3. What is $hao$  in line 169?
>
> Sequences of actions and observations to time $t$ are maintained as histories $h_{t} = [a_{0}, o_{0}, ... , a_{t}, o_{t}]$. $h$ is the root node, $ha$ is the node below for the action taken, and $hao$ is the subsequent node for the observation received. In Figure 1, the $h$ and $hao$ nodes are circles and $ha$ nodes are squares.
>
> 4. Shouldn’t the probabilities in the last column of Table 1 sum to 1?
>
> If all the actions were included in the analysis, the columns in Table 1 would sum to one. Table 2 shows Constrained LightDark has 7 discrete actions. The only actions included in Table 1 are +1, +5, and +10 because they are reasonable initial actions to locate the agent from the starting position by moving towards, and not away from, the light region. We want to compare how often the differing methods select the optimal action (+5) of those 3 reasonable actions for their first step.
>
> [1] Poupart, P.; Malhotra, A.; Pei, P.; Kim , K.-E.; Goh, B .; and Bowling , M. 2015. Approximate linear programming for constrained partially observable Markov decision processes. 325 In AAAI Conference on Artificial Intelligence (AAAI) , volume 29.

---

### Meta-Review · Area_Chair_8CU1 · 2024-02-02

**Recommendation:** Accept (Oral)
**Confidence:** 4

**Metareview:**

The reviewers agree that this paper makes an interesting and valuable contribution to solving Constrained POMDPs.

We ask the authors to follow the reviewers' recommendations and include additional clarifications to improve the clarity and presentation of the paper.

**Ethical Considerations:**

(1) Not Applicable: The paper does not have any ethical considerations to address